# Volumetric Changes after Coblation Ablation Tongue (CAT) in Obstructive Sleep Apnea Patients

**DOI:** 10.3390/jcm11144186

**Published:** 2022-07-19

**Authors:** Yi-An Lu, Chao-Jan Wang, Yen-Ting Chiang, Hsueh-Yu Li

**Affiliations:** 1Department of Otolaryngology Head and Neck Surgery, Linkou Chang Gung Memorial Hospital, Taoyuan 333, Taiwan; season4990@hotmail.com (Y.-A.L.); rita10071@gmail.com (Y.-T.C.); 2School of Medicine, Chang Gung University, Taoyuan 333, Taiwan; cjwang@cgmh.org.tw; 3Department of Medical Imaging and Intervention, Linkou Chang Gung Memorial Hospital, Taoyuan 333, Taiwan

**Keywords:** coblation ablation tongue (CAT), tongue volume, tongue length, computed tomography, obstructive sleep apnea

## Abstract

Background: Obstruction of the tongue is commonly seen in patients with obstructive sleep apnea (OSA). This study proposed whole tongue treatment using coblation ablation tongue (CAT) and aimed to explore the potential association between the dimensions of a tongue and the severity of OSA, to inspect volumetric changes of the tongue after CAT, and to search for factors that influence outcome of tongue volume change. Methods: The prospective study enrolled 12 OSA patients (all male, average age: 35 years, average apnea/hypopnea index (AHI): 45.5 event/h, average body mass index (BMI): 27.0 kg/m^2^). All patients received multi-level sleep surgery including septomeatoplasty, uvulopalatopharyngoplasty, and CAT. The CAT used a coblation wand to perform uniform multiple ablations (15 points, body −6, base −9) on dorsal tongue. Three dimensions of the tongue (length, height, and width) and tongue volume were measured from head and neck computed tomography. The perioperative changes in the tongue dimension/volume and AHI were assessed at baseline and 3 months after surgery. Result: The baseline tongue length and AHI had a significant correlation (r = 0.60, *p* = 0.02). The multi-level surgery significantly improved AHI (43.8 vs. 23.7, *p* = 0.008). The CAT significantly decreased tongue volume from 91.3 to 85.6 cm^3^ (*p* = 0.02), with an average tongue volume reduction of 5.7 cm^3^ per person and 0.38 cm^3^ per ablation. Further outcome analysis showed surgical success was significantly higher in patients with non-hypertrophic lingual tonsils (grade I/II) than in those with hypertrophic lingual tonsils (grade III/IV) (*p* = 0.02). Conclusion: Length of the tongue is associated with the severity of OSA. The CAT significantly decreased the tongue volume in OSA patients. A volumetric reduction of 0.38 cm^3^ per ablation could be useful in the optimal reduction of tongue for OSA. The CAT significantly enlarged the retroglossal airway volume, which is related to the non-hypertrophic lingual tonsil.

## 1. Introduction

For patients with obstructive sleep apnea (OSA), continuous positive airway pressure (CPAP) is considered the first-line and gold standard treatment [1]. However, for various reasons, many people are intolerant or unwilling to receive CPAP therapy and seek surgical intervention as salvage or alternative treatment [2]. Among the sleep surgeries, uvulopalatopharyngoplasty (UPPP) is the most commonly used procedure for treating snoring and OSA [3]. However, the low success rate of UPPP for OSA has been criticized for decades [4]. Many studies investigating the outcomes of UPPP showed that persistent tongue obstruction was the main cause of UPPP failure [5,6].

Volume reduction in the tongue in multi-level surgery is often added to OSA surgery to improve its success rate [7,8,9]. Two meta-analysis studies of tongue volume reduction showed that radiofrequency tissue ablation (RFA) at the base of the tongue in multi-level procedures is a clinically effective tool for the reduction in the respiratory disturbance index (RDI) levels of OSA patients [10,11]. However, previous studies focused mainly on volume reduction in the tongue base and the change in adverse sleep parameters after RFA [10,11], with no discussions on the association of OSA and individual tongue parameters, the amount of tongue volume reduction in single ablation, and the difference in tongue volume reduction between surgical success and failure group patients and why.

This study advocates a mini-invasive model of whole tongue treatment including the tongue body and tongue base via coblation, involving the measurement of the perioperative changes in the volume of the tongue and airspace using three-dimensional computed tomography (CT) scan in OSA patients. The aims of this study were as follows: (1) To identify the exact association between disease severity of OSA in terms of apnea/hypopnea index (AHI) and individual tongue parameter (length, height, and width); (2) To measure the average tongue volume reduction in each patient and in one single ablation; (3) To compare the difference in tongue volume reduction between the surgical success and failure groups.

## 2. Methods

### 2.1. Ethics Statement

This prospective study was supported by Chang Gung Medical Foundation grants (CMRPG3K0271) and approved by the Institutional Review Board of the Chang Gung Memorial Foundation (number: 201901113A3). Linkou Chang Gung Memorial Hospital is the main branch of Chang Gung Memorial Hospitals. The IRB of the Chang Gung Medical Foundation is the representative and in charge of all branches of the Chang Gung Memorial Hospital in IRB review affairs. Informed consent was obtained from all participants.

### 2.2. Study Population

Twelve OSA patients were enrolled in this study. The inclusion criteria were an age between 20 and 65 years, a body mass index (BMI) of < 32 kg/m^2^, AHI > 15 event/h, deviation of nasal septum in nasal examination, collapse of the palate and tongue during drug-induced sleep endoscopy [12]. The exclusion criteria were significant retrognathia or syndromic patients, co-morbidity with severe medical disease, previous palate or tongue surgery, and a high risk for general anesthesia (American Society of Anesthesiologists physical status class > 2). Given that we aimed to understand the effect of CAT on OSA, patients with palatal tonsil hypertrophy (grade 3, 4) were also excluded to minimize the influence of tonsillectomy. Patients were intolerant or unwilling to use CPAP therapy and sought surgical intervention as an alternative salvage treatment. All patients received multi-level upper airway surgery that included septomeatoplasty (SMP), UPPP, and coblation ablation tongue (CAT) based on the findings observed on physical examination and during drug-induced sleep endoscopy (DISE). Independent DISE was performed in surgical OSA patients before any operation. The procedure of DISE was implemented as follows: an A-2000 bispectral index Vista monitor (version 3.11; Aspect Medical Systems, Inc., Newton, MA, USA) was used to monitor the depth of sedation. Intravenous propofol (10 mg/mL; AstraZeneca, Caponago, Italy) was initially administered at 0.5 mg/kg, and further doses of 10 to 20 mg were given every 30 s to achieve the target level of light sedation (bispectral index, 70–75) for endoscopic examination. Surgical success was defined as Sher’s criteria [13], with a reduction from a pre-surgery AHI of at least 50% and a post-surgery AHI of less than 20 per hour.

### 2.3. Polysomnography

Level I PSG (Nicolet UltraSom System, Madison, WI, USA) was performed to document sleep and breathing in all patients. The main PSG parameter used in this study was AHI. Apnea was defined as a drop in the peak thermal sensor excursion by at least 90% of baseline for at least 10 s. Hypopnea was defined as a decrease in airflow accompanied by desaturation of 4% oxygen [14]. The AHI was defined as the number of total apneic and hypopneic episodes per hour of sleep. The polysomnographic data were scored by one experienced sleep specialist who was blind to the image data. 

### 2.4. 3-D CT Measuring Algorithm

The CT images were obtained using an Aquilion One system (320-detector row, Toshiba, Japan), using a dynamic volume scan protocol when the patients were awake, in a supine position, and at end-expiratory status. Then, the images were analyzed and reconstructed by post processing software (Virtual Place; AZE Inc., Tokyo, Japan). Three diameters of the tongue component (length, height, and width) and tongue volume were calculated. Tongue length is the anterior–posterior length of the tongue and was measured from the upper incisor to the upper margin of the epiglottis in the sagittal view of the CT scan. Tongue height was measured as the line vertical to the tongue length and crossing the midline of the soft palate (Figure 1A). Tongue width was measured as the maximal lateral diameter of the tongue in axial sections (Figure 1B). The tongue volume was calculated as the sum of all cross-sectional areas of the tongue. We also analyzed the cross-sectional area and volume of the airway at the retropalatal (RP) and retroglossal (RG) levels and the total airway volume. The RP area was defined as the minimal cross-sectional area from the hard palate to the distal end of the uvula. The RP volume was defined as the air volume between the hard palate and distal end of the uvula. The RG area was defined as the minimal cross-sectional area from the distal end of the uvula to the upper margin of the epiglottis. The RG volume was defined as the air volume between the distal end of the uvula and the upper margin of the epiglottis. The total volume was the sum of the cross-sectional area from the hard palate to the upper margin of the epiglottis (Figure 2). One experienced radiologist (Wang CJ), who was blind to polysomnographic data, analyzed the images. 

### 2.5. Surgical Plan and Design for Coblation Whole Tongue Surgery

The CAT is characterized by its minimal invasiveness (mucosa preservation, needle-hole wound), hybrid technique (muscle scarring/contraction, fat ablation), and whole tongue (body and base) treatment (Figure 3). The coblation wand (Reflex 4855, Anthrocare Corp., Austin, TX, USA) was used with the ablation-5 mode to insert into the tongue, with a continuous energy release of 15 s for volume reduction, followed by a coagulation-4 mode for precautionary hemostasis during withdrawal of the wand. The operation was initiated by traction suture using a 3-0 silk through the anterior midline tongue. The midline of the dorsal tongue and circumvallate papillae were marked as a reverse T-shape in a blue color. The CAT was implemented at the dorsal tongue. The needle of the wand was inserted fully into the tongue, with an active zone 1 cm away from the mucosa to avoid thermal damage to the taste buds. Procedures of CAT included 6-point ablation of the tongue body and 9-point ablation of the tongue base (Figure 4). Postoperative care involved humidified oxygen support, positional therapy (elevation of the head of bed), prophylactic systemic antibiotic (Cefazolin 1 gm g6h), intravenous analgesia (Ketololac 30 mg g6h), and intravenous dexamethasone (5 mg, q6h) for 3 days.

### 2.6. Statistical Analysis

Statistical analysis was performed using SPSS for Mac 21.0 (Statistical Package for Social Sciences; SPSS Inc., Chicago, IL, USA). Data are expressed as the mean ± standard deviation (S.D.). A Mann–Whitney U test was used for comparing the non-parametric data, and the X^2^ test was used for comparing the categorical data between the groups. Wilcoxon signed rank test were used to compare the pre- and post-operative data. Spearman correlation analysis was used to evaluate the relationships between variables. Significance was set as <0.05.

## 3. Results

Among the 12 male OSA patients, the average age was 34.9 ± 3.2 years old. The average preoperative AHI was 45.5 ± 9.1/hour, and BMI was 27.0 ± 1.9 kg/m^2^. The assessment comprised of polysomnography and upper airway CT at baseline and 3 months after treatment. The association between the CT parameters and OSA severity showed that tongue length, instead of width or height, was the only parameter of tongue component that correlated significantly with AHI (r = 0.60, *p* = 0.02) (Table 1). After multi-level surgery, AHI decreased from 43.8 ± 16.9 to 23.7 ± 13.4/hour (*p* = 0.008) with surgical success in eight patients (66.6%). Subjective daytime sleepiness survey in terms of ESS also improved from 12.5 ± 5.1 to 7.3 ± 2.6 (*p* = 0.005) (Table 2). The tongue volume reduced significantly from 91.3 ± 12.6 to 85.6 ± 10.4 cm^3^ (*p* = 0.02), with an average tongue volume reduction of 5.7 cm^3^ per person and 0.38 cm^3^ per ablation. Changes of tongue volume were insignificantly correlated with changes of AHI (r = 0.21, *p* = 0.50). Comparing preoperative and postoperative changes in the success and failure groups showed that the RG area, RG volume, and total volume were significantly larger in the success group than in the failure group (*p* = 0.04, 0.001, and 0.02, respectively) (Table 3). Further outcome analysis based on lingual tonsil grading [15], revealed that all patients in the success group had non-hypertrophic lingual tonsils (Gr I/II, *n* = 8). By contrast, three patients in failure group had hypertrophic lingual tonsils (Gr III/IV), and only one patient had a Gr I lingual tonsil. There was a significant difference (*p* = 0.02) between lingual tonsil grade (Gr I/II vs Gr III/IV) and surgical outcome (success vs failure).

## 4. Discussion

Volumetric changes can be measured by CT or magnetic resonance imaging (MRI). Although MRI provides a better resolution in soft tissue than CT, it is expensive and needs a long queue. By contrast, CT scans are less expensive and more compliant with the implementation of the study. To the best of our knowledge, this study is the first to comprehensively assess the association between the severity of OSA and the three major dimensions of the tongue, the effect of tongue volume reduction per ablation, and the cause of perioperative changes in volume/airspace between the success and failure groups. The results showed that the length, instead of the height, of the tongue was correlated with the AHI. The average CAT-producing tongue volume reduction was 5.7 cm^3^ per patient and 0.38 cm^3^ per ablation. In the success group, the RG area and RG volume were significantly wider than those in the failure group, and this was related to the non-hypertrophy (grade I/II) of lingual tonsils.

The use of images to investigate the association between the airspace and AHI has been widely documented [16]. Most studies have shown that the lateral dimension of the retropalatal area is related to OSA severity in terms of the AHI [17,18]. Furthermore, the volume of the tongue is also positively correlated with the AHI [19,20]. However, no study has probed into detailed relationship between AHI and any individual dimension of the tongue. In this study, we measured the three major dimensions of the tongue in terms of length, height, and width and compared them with the AHI. The results showed that length was the only tongue dimension associated with the severity of OSA. This makes sense because the direction of tongue collapse emerges from anterior to posterior during drug-induced sleep endoscopy and drug-induced sleep imaging [21]. The increased length of the tongue contributes to narrower retroglossal airspace and is more prone to collapse during supine sleep. In the meanwhile, this suggests that OSA with tongue obstruction can be improved by shortening the length of tongue, which can be achieved by tensing the tongue muscle, ablating the tongue fat, and performing a resection the lingual tonsils.

Minimally invasive volume reduction in the tongue by RFA was first demonstrated in an animal model by Powell et al. in 1997 [22]. The RFA can be absorbed in deep tissue at relatively low temperatures, offering the advantages of less pain and less morbidity than conventional excisional surgery [10]. However, the effect of tongue volume reduction after RFA is not clear. Although CT analysis is a validated tool to measure the tongue and upper airway volume in OSA [23,24], most previous studies compared the difference in tongue volume and airway space between OSA and normal subjects without measuring the volumetric changes after RFA [25,26]. Previous studies used radiofrequency for tongue volume reduction. Only one abstract [27] and one article [28] mentioned the use of coblation for tongue volume reduction with the heading of the coblation channeling tongue. In this study, coblation was endowed with new territory (the whole tongue instead of the tongue base) and a new task (fat ablation and muscle contraction instead of channeling). Therefore, we give this mini-invasive procedure a new heading—CAT. Our study is the first to describe the volumetric changes of the tongue after CAT. In this study, tongue volume was reduced significantly after CAT, with an average volume reduction of 5.7 cm^3^ per patient and 0.38 cm^3^ per ablation. In OSA surgery, adequate volume reduction in the tongue is crucial for a good surgical outcome. From the outcome survey in transoral robotic surgery, a tongue volume reduction of 7 cm^3^ (mL) is the cutoff point between optimal and suboptimal surgical results [29]. Consequently, it is suggested to implement CAT of more than 18 ablations for optimal tongue volume reduction in OSA patients. Although there was no statistical significance, there was a trend towards the increase in retroglossal area and volume after CAT. The authors that speculated open-mouth breathing during CT scans in some patients confounded the data, since breathing through the mouth is associated with reduction in the retropalatal and retroglossal area of the upper airway in patients with OSA [30]. 

For OSA outcomes, a meta-analysis of 16 studies revealed that RFA is effective in reducing the RDI in OSA patients 10. Extended meta-analysis also showed that RFA of the tongue base reduced RDI in a multi-level procedure [11]. In this study, the AHI improved from 43.8 to 23.7 (*p* < 0.01), and the ESS improved from 12.5 to 7.5 (*p* < 0.01) after multi-level surgery including CAT. The exact AHI reduction specifically from CAT in simultaneous multi-level surgery is unclear. However, the RG area (161.0 vs. 73.8, *p* = 0.004) and RG volume (11.1 vs. 6.3, *p* = 0.001) were significantly larger in the success group than in the failure group. This suggests that optimal RG airspace from effective tongue volume reduction contributes to surgical success. Further outcome analysis revealed that there was a significant difference between lingual tonsil grade and surgical outcome (*p* = 0.02). All success patients had non-hypertrophic lingual tonsil (Gr I/II), by contrast, seventy-five percent of failure patients had hypertrophic lingual tonsils (Gr III/IV). This suggests that CAT is less effective in reducing tongue volume in which hypertrophic lingual tonsils (grade III/IV) are composed, and combined lingual tonsillotomy is necessary from the viewpoint of volumetric reduction and surgical outcomes. However, we presume that CAT can still be concurrently implemented because of its irreplaceable role in tensing the muscle and reducing intramuscular fat in hypertrophic lingual tonsil group patients. It is noteworthy that one patient in the failure group had non-hypertrophic lingual tonsil (grade I) who was further assessed for muscle strength and endurance of the tongue by the Iowa oropharyngeal performance instrument, and that the result showed normal strength and abnormally low endurance (posterior). This implies that low muscle tone of the tongue can be another factor contributing to failure of tongue volume reduction surgery for OSA. The study focused on the volumetric change of the tongue following the mini-invasive procedure instead of AHI and desaturation. Therefore, smoking and other comorbidities as the potential confounding factors were not investigated. However, we determined that long-term smoking is related to postoperative taste disturbance in our previous research, and that is necessary to inform the OSA patients of this before intrapharyngeal surgery.

This study has several limitations. Firstly, it is a small sample size study with moderate to severe OSA patients, which may not be generalizable to a larger population. The sample size of the study was limited because of patients’ concern about the radiation dose and the expense of the CT scans depleting the study budget. Extended study to recruit more patients or a meta-analysis of similar studies may improve the power of the evidence. Secondarily, the performance of simultaneous multi-level surgery for OSA obfuscated the effect of CAT on AHI. Future studies through staged operation or with a control group can help elucidate the individual contribution of CAT in ameliorating the severity of OSA. Thirdly, long-term volumetric change is unknown. A three month follow-up CT scan was adopted to meet the timeframe of a one year study. Based on the results, an extended follow-up is ongoing to observe the long-term volumetric changes of the tongue and the timing of salvage treatments for relapsing patients.

## 5. Conclusions

Length of the tongue is associated with the severity of OSA. A potential strategy in treating OSA with tongue obstruction emerges from this observation. In other words, we may need to shorten the length in order to cause a decrease in AHI. The CAT process significantly decreases the volume of tongue, which indicates the feasibility of this mini-invasive tongue procedure. A volumetric reduction of 0.38 cm^3^ per ablation could be valuable information for surgical plans in the optimal reduction in tongue volume via the mini-invasive procedure in OSA patients. To put it another way, some suboptimal results in volumetric change may be attributed to insufficient ablations. For overweight OSA patients with tongue obstruction, 20 ablations are suggested. The CAT process significantly enlarged the retroglossal airway volume in the success group, which is related to non-hypertrophic lingual tonsils. For OSA patients with tongue obstruction and hypertrophic lingual tonsils, standalone CAT is insufficient in terms of volume reduction in the tongue, and an additional lingual tonsillotomy is suggested. Our small sample size limits the power of interpretation in this study, and further research is warranted.

## Figures and Tables

**Figure 1 jcm-11-04186-f001:**
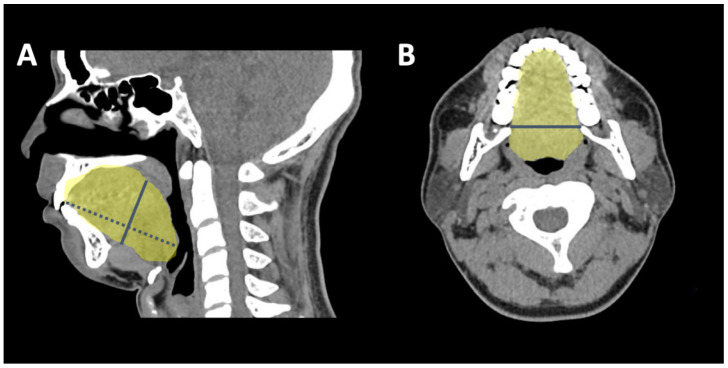
(**A**,**B**): Tongue length is the anterior–posterior length of the tongue and was measured from the upper incisor to the upper margin of the epiglottis in the sagittal view of the computed tomography scan. Tongue height was measured as a line vertical to the tongue length and crossing the midline of the soft palate (**A**). Tongue width was measured as the maximal lateral diameter of the tongue in the axial section (**B**).

**Figure 2 jcm-11-04186-f002:**
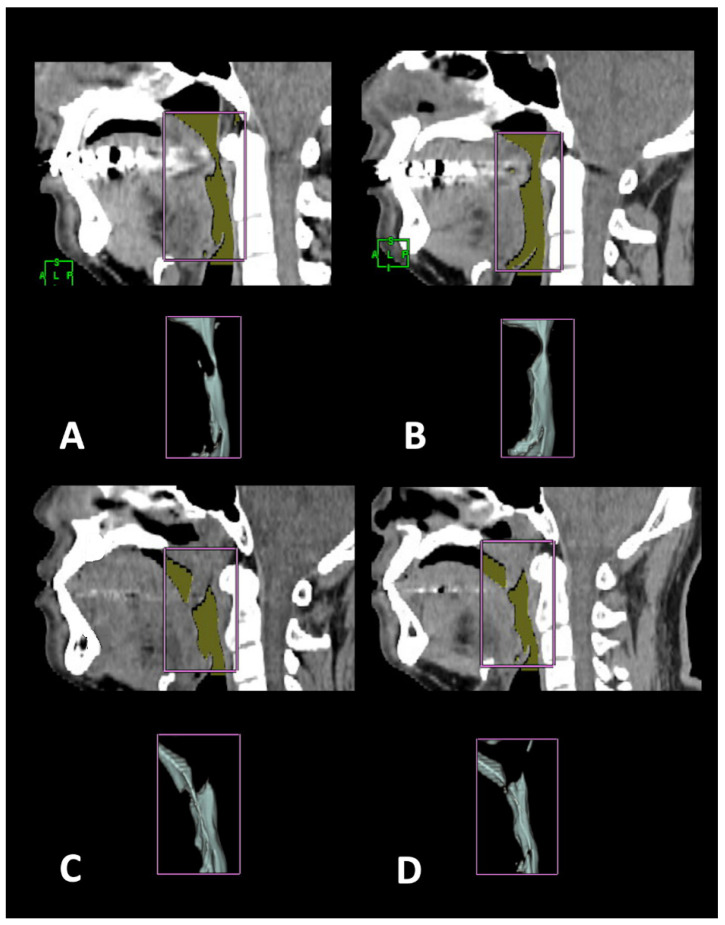
Total airway volume in one surgical success patient: (**A**) pre-surgery, (**B**) post-surgery. Total airway volume in one surgical failure patient: (**C**) pre-surgery, (**D**) post-surgery.

**Figure 3 jcm-11-04186-f003:**
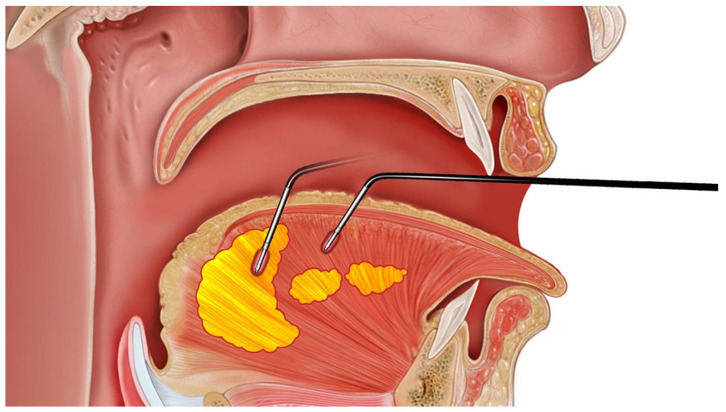
The coblation ablation tongue (CAT) is characterized by its hybrid technique (muscle scarring/contraction, fat ablation) and whole tongue (body and base) treatment.

**Figure 4 jcm-11-04186-f004:**
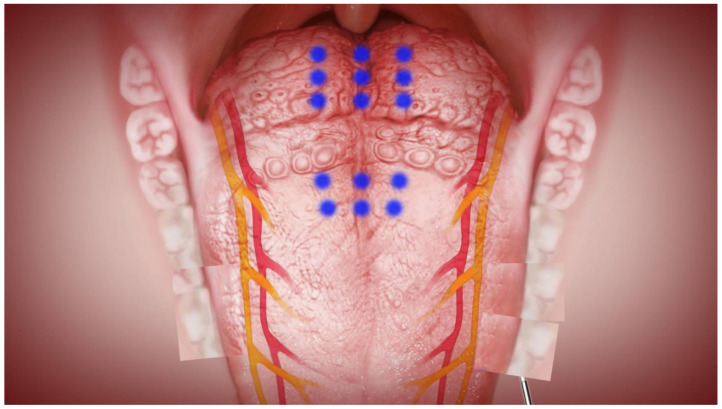
Procedures of the coblation ablation tongue (CAT) included 6-point ablation of the tongue body and 9-point ablation of the tongue base.

**Table 1 jcm-11-04186-t001:** Spearman correlation to baseline AHI.

Parameter	r Value	*p* Value
Basic assessment		
BMI (kg/m^2^)	0.61	0.02 *
Age (year)	0.23	0.41
Airway minimal area and diameter		
RP area (cm^2^)	−0.19	0.49
RG area (cm^2^)	−0.01	0.99
Airway volume		
Total volume (cm^3^)	−0.33	0.23
RP volume (cm^3^)	0.42	0.12
RG volume (cm^3^)	−0.43	0.11
Tongue volume and diameter		
Tongue volume (cm^3^)	0.35	0.20
Tongue length (cm)	0.60	0.02 *
Tongue height (cm)	−0.03	0.90
Tongue width (cm)	0.91	0.26

* *p* < 0.05; here, AHI: apnea-hypopnea index; BMI: body mass index; RP: retropharyngeal; RG: retroglossal.

**Table 2 jcm-11-04186-t002:** Changes in the parameters after surgery.

	Pre-Surgery	Post-Surgery	
Parameter	Mean ± Std.	Mean ± Std.	*p* Value
Basic assessment			
AHI (event/h)	43.8 ± 16.9	23.7 ± 13.4	0.008 **
AI (event/h)	28.4 ± 15.6	11.6 ± 16.8	0.01 *
ESS (0–24)	12.5 ± 5.1	7.3 ± 2.6	0.005 **
Airway minimal area and diameter			
RP area (cm^2^)	27.4 ± 26.7	32.7 ± 44.0	0.66
RG area (cm^2^)	96.2 ± 63.7	131.9 ± 72.9	0.12
Airway volume			
Total volume (cm^3^)	11.6 ± 5.3	12.9 ± 3.3	0.21
RP volume (cm^3^)	2.7 ± 1.9	2.8 ± 2.4	0.88
RG volume (cm^3^)	8.2 ± 4.2	9.5 ± 2.8	0.24
Tongue volume and diameter			
Tongue volume (cm^3^)	91.3 ± 12.6	85.6 ± 10.4	0.02 *
Tongue length (cm)	71.8 ± 6.4	71.2 ± 6.3	0.46
Tongue height (cm)	43.4 ± 6.5	44.5 ± 5.9	0.21
Tongue width (cm)	46.5 ± 3.6	45.2 ± 5.0	0.70

Here, AHI: apnea-hypopnea index; AI: apnea index; ESS: Epworth sleepiness scale; RP: retropharyngeal; RG: retroglossal; * *p* < 0.05; ** *p* < 0.01.

**Table 3 jcm-11-04186-t003:** The success group versus the failure group.

	Pre-Surgery			Post-Surgery		
Parameter	Success Group (*n* = 8)	Failure Group (*n* = 4)	*p* Value	Success Group (*n* = 8)	Failure Group (*n* = 4)	*p* Value
Airway minimal area						
RP area (cm^2^)	31.3 ± 27.2	19.5 ± 27.8	0.50	44.3 ± 50.4	9.5 ± 10.2	0.21
RG area (cm^2^)	99.3 ± 54.8	90.0 ± 88.3	0.83	161.0 ± 70.4	73.8 ± 33.9	0.04 *
Airway volume						
Total volume (cm^3^)	12.4 ± 4.0	10.2 ± 7.9	0.53	14.4 ± 3.0	7.0 ± 1.2	0.02 *
RP volume (cm^3^)	2.9 ± 2.3	2.0 ± 0.8	0.50	2.8 ± 2.4	2.1 ± 0.6	0.61
RG volume (cm^3^)	9.1 ± 3.8	6.5 ± 5.0	0.32	11.1 ± 1.8	6.3 ± 1.2	0.001 **
Tongue volume and diameter						
Tongue volume (cm^3^)	89.3 ± 11.2	95.4 ± 16.0	0.45	82.1 ± 8.4	92.8 ± 11.5	0.09
Tongue length (cm)	70.0 ± 5.9	75.4 ± 6.4	0.18	68.8 ± 5.0	76.1 ± 6.2	0.05
Tongue height (cm)	44.6 ± 6.6	41.0 ± 6.6	0.39	45.1 ± 6.7	43.2 ± 4.4	0.62
Tongue width (cm)	46.7 ± 4.2	46.1 ± 2.3	0.79	43.5 ± 4.8	48.7 ± 3.8	0.09

Here, RP: retropharyngeal; RG: retroglossal; * *p* < 0.05; ** *p* < 0.01.

## Data Availability

The data are not publicly available due to the regulation of our institution and protection of patients’ privacy in this small sample size group. However, the data presented in this study are available on request from the corresponding author for further research, if available.

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
