# Peer review of "Volumetric Changes after Coblation Ablation Tongue (CAT) in Obstructive Sleep Apnea Patients"

_jcm, 2022, doi:10.3390/jcm11144186_

Round 1

Reviewer 1 Report

The authors aim to demonstrate: (1) the association between severity in OSA patients and individual tongue parameters such as length, height, width, (2) to measure the average tongue volume reduction, (3) to compare the differences in tongue volume reduction between the success and failure group.

There are a few concerns which need to be addressed: 

1. The authors need to describe the methods and findings systematically according to the aim. 

2. What was the DISE protocol used

3. How were the confounding factors addressed such as smokers, other comorbid

4. Provide reference for CAT method used

5. What was the intra-op and post-op protocol? Are any dexamethosone/antibiotics prescribed?

6. What are the other modes for volumetric changes measurement? Please elaborate on this and why was CT used?

7. Why were subjective parameters not included? such as ESS/ OSA-18 tool

Author Response

Dear editor and reviewers

Thank you so much for providing us plenty of informative and constructive comments that are indispensable to the integrity of this article.

The authors have responded to individual comments point-by-point and amended accordingly in the text (marked by yellow color).

Hopefully the manuscript is closer to your consideration of publication.

  1. The authors need to describe the methods and findings systematically according to the aim

Reply: Thanks for the great comment.

  • “This study proposed whole tongue treatment by coblation ablation tongue (CAT) and aimed to explore the potential association between dimensions of tongue and severity of OSA, inspect volumetric changes of tongue after CAT, and search for factors that influence outcome of tongue volume change.
  • We revised the aims in the background of Abstract to comply with the methods and findings (results) systemically.
  •  
  1. What was the DISE protocol used

Reply: Thanks for the important comment.

  • Independent DISE was performed in surgical OSA patients before operation. The procedure of DISE was implemented as following: An A-2000 bispectral index Vista monitor (version 3.11; Aspect Medical Systems, Inc, Newton, Massachusetts) was used to monitor the depth of sedation. Intravenous propofol (10 mg/mL; AstraZeneca, Caponago, Italy) was initially administered at 0.5 mg/kg, and further doses of 10 to 20 mg were given every 30 seconds to achieve the target level of light sedation (bispectral index, 70-75) for endoscopic examination.
  • We added this in the Method
  •  
  1. How were the confounding factors addressed such as smokers, other comorbid

Reply: Thanks for the remarkable comment.

  • “The study focused on the volumetric change of tongue following the mini-invasive procedure instead of AHI and desaturation. Therefore, smoking and other comorbid as the potential confounding factors were not investigated. However, we found long-term smoking is related to postoperative taste disturbance in our previous research that is necessary to inform the OSA patients before intrapharyngeal surgery. “
  • We added this paragraph in Discussion

  1. Provide reference for CAT method used

Reply: Thank you for this important comment

  • Previous studies used radiofrequency for tongue volume reduction. Only one article and one abstract mentioned the use of coblation for tongue volume reduction with the heading of coblation channeling tongue. In this study, coblation was endowed with new territory (whole tongue instead of tongue base) and new task (fat-ablation & muscle-contraction instead of channeling). Therefore, we give the mini-invasive procedure a new heading-CAT.
  • We added the two articles in the Reference

  1. What was the intra-op and post-op protocol? Are any examethosone/antibiotics prescribed?

Reply: Thanks for the reminding

  • Postoperative care involved humidified oxygen support, positional therapy (elevation of the head of bed), prophylactic systemic antibiotic (Cefazolin 1gm g6h), intravenous analgesia (Ketololac 30mg g6h), and intravenous dexamethasone (5mg, q6h) for 3 days.
  • We add these in the Methods

  1. What are the other modes for volumetric changes measurement? Please elaborate on this and why was CT used?

Reply: Thanks for the enquiry.

  • Volumetric changes can be measured by CT or MRI. MRI provides a better resolution in soft tissue than CT. However, it’s expensive and needs a long queue. By contrast, CT scan is less expensive and more compliant with the implementation of the study.
  • We added these in the Discussion

  1. Why were subjective parameters not included? such as ESS/ OSA-18 tool

Reply: Thanks for good comment.

  • “Subjective daytime sleepiness survey in terms of ESS also improved from 12.5±5.1 to 7.3±2.6 (p=0.005).
  • We amended this in the Results

Thank you for your highly professional comments

Reviewer 2 Report

This study is interesting and well designed, but as the authors point out, it has three major limitations:

This study has several limitations. Firstly, it’s a small sample size study with moderate to severe OSA patients, which may not be generalizable to a larger population.

Secondarily, the performance of simultaneous multilevel surgery for OSA obfuscated the effect of CAT on AHI.

Thirdly, long-term volumetric change is unknown.

In addition, it refers to the possible nasal obstruction due to deviation of the nasal septum but no functional study is recorded that would allow it to be confirmed, such as active anterior rhinomanometry.

In my opinion, the study is interesting, but I do not believe that it provides relevant evidence if the limitations indicated are not overcome.

Author Response

Dear editor and reviewers

Thank you so much for providing us plenty of informative and constructive comments that are indispensable to the integrity of this article.

The authors have responded to individual comments point-by-point and amended accordingly in the text (marked by yellow color).

Hopefully the manuscript is closer to your consideration of publication.

  1. it’s a small sample size study with moderate to severe OSA patients, which may not be generalizable to a larger population.

Reply: thanks for reminding

It is genuine limitation for this study that we have claimed in the limitation (Discussion) and stated the whys and wherefores of the situation.

It’s a small sample size study with moderate to severe OSA patients, which may not be generalizable to a larger population. The sample size of the study was limited because of patient’s concern about the radiation dose and the expense of the CT scans depleting the study budget. Extended study to recruit more patients or meta-analysis of similar studies may improve the power of evidence. (Limitation, Discussion)

  1. The performance of simultaneous multilevel surgery for OSA obfuscated the effect of CAT on AHI.

Reply: thanks for the criticism

This is the real situation that the performance of a simultaneous multi-level surgery for OSA obfuscated the effect of CAT on AHI. The authors have proposed two extended methods to elucidate the effect of CAT on AHI by stage operation or with control group. (Limitation, Discussion)

3.Long-term volumetric change is unknown.

Reply: Thank you for this important comment

It’s always difficult to long-term follow up surgical OSA patients since they are unwilling to come back if their clinical symptoms of OSA had improved (majority). In our previous study, only 5% of surgical OSA patients were willing to return for a 3-year follow up. It’s noteworthy that all of them were symptomatic with persistent /recurrent snoring or daytime sleepiness. This reflects the potential selection bias of long term follow up in surgical OSA patients. However, we will continue following up this group patients and provide salvage treatment for relapsing patients. (Limitation, Discussion)

4.It refers to the possible nasal obstruction due to deviation of the nasal septum but no functional study is recorded that would allow it to be confirmed, such as active anterior rhinomanometry.

Reply: Thank you for the influential comment

Nasal obstruction is an important factor with aggravation of snoring and OSA. Accordingly, SMP is a routine procedure in multi-level surgery for OSA patients with clinical symptoms of daytime/nocturnal nasal obstruction, mouth breathing and septal deviation in nasal examination. We do not perform rhinomanometry in every surgical patient with nasal obstruction. However, we have some related reports on this issue for your reference:

  • Li HY, Wang PC, Hsu CY, Cheng ML, Liou CC, Chen NH. Nasal resistance in patients with obstructive sleep apnea. ORL 2005; 67:70-74.
  • Li HY, Engleman H, Hsu CY, Izci B, Vennelle M, Cross M, Douglas NJ. Acoustic reflection for nasal airway measurement in patients with obstructive sleep apnea/hypopnea syndrome. Sleep 2005; 28:1554-1559.
  • Li HY, Lee LA, Wang PC, Chen NH, Lin Y, Fang TJ. Nasal surgery for snoring in patients with obstructive sleep apnea. Laryngoscope 2008; 118: 354-359.
  • Li HY, Lin Y, Chen NH, Lee LA, Fang TJ, Wang PC. Improve in quality of life after nasal surgery for patients with obstruction sleep apnea and nasal obstruction. Arch Otolaryngol Head Neck Surg 2008; 134(4): 429-433.
  • Li HY, Lee LA, PC Wang, Fang TJ, Chen NH. Can nasal surgery improve obstructive sleep apnea: subjective or objective? Am J Rhinol Allergy 2009; 23(6): e51-e55.
  • Li HY, Pa-Chun Wang, Yu-Pin Chen, LA Lee, TJ Fang, HC Lin. Critical appraisal and meta-analysis of nasal surgery for obstructive sleep apnea. Am J Rhinol Allergy. 2011; 25(1):45-49.
  • Tsai MS, Chen HC, Liu SY, Lee LA, Lin CY, Chang GH, Tsai YT, Lee YC, Hsu CM, Li HY*. Holistic care for obstructive sleep apnea (OSA) with an emphasis on restoring nasal breathing: A review and perspective. J Chin Med Assoc. 2022; 85(6): 672-678.

  1. In my opinion, the study is interesting, but I do not believe that it provides relevant evidence if the limitations indicated are not overcome.

Reply: Thank you for the influential comment. Although some limitations, this study offers some novelties and niches for sleep surgery:

  1. Length of the tongue is associated with severity of OSA that emerges a potential strategy in treating OSA with tongue obstruction. In other words, we may need to shorten the length in terms of decrease of AHI.
  2. Volumetric reduction of 0.38 cm3 per ablation could be valuable information for surgical plan in optimal reduction of tongue volume via mini-invasive procedure in OSA patients. To put it another way, some suboptimal results in volumetric change may be attributed to insufficient ablations. For over-weighted OSA patients with tongue obstruction, 20 ablations are suggested.
  3. CAT significantly enlarged the retroglossal airway volume in success group, which is related to non-hypertrophic lingual tonsil. For OSA patients with tongue obstruction and hypertrophic lingual tonsil, stand-alone CAT is insufficient in volume reduction of the tongue, and additional lingual tonsillotomy suggested. (Conclusion)

The authors hope these points will be helpful for surgical decision-making and further research.

Once again, thank you for the highly professional comments

Round 2

Reviewer 1 Report

The authors have addressed all the comments adequately.